# Offline Reinforcement Learning Under Value and Density-Ratio Realizability: The Power of Gaps

**Jinglin Chen, Nan Jiang**[1]

[1]Department of Computer Science, University of Illinois Urbana-Champaign, Urbana, IL, USA

## Abstract

We consider a challenging theoretical problem in offline reinforcement learning (RL): obtaining sample-efficiency guarantees with a dataset lacking sufficient coverage, under only realizability-type assumptions for the function approximators. While the existing theory has addressed learning under realizability and under non-exploratory data separately, no work has been able to address both simultaneously (except for a concurrent work which we compare in detail). Under an additional gap assumption, we provide guarantees to a simple pessimistic algorithm based on a version space formed by marginalized importance sampling (MIS), and the guarantee only requires the data to cover the optimal policy and the function classes to realize the optimal value and density-ratio functions. While similar gap assumptions have been used in other areas of RL theory, our work is the first to identify the utility and the novel mechanism of gap assumptions in offline RL with weak function approximation.

## 1 INTRODUCTION AND RELATED WORKS

In offline reinforcement learning (RL), the learner searches for a good policy purely from historical (or offline) data, without direct interactions with the real environment. The lack of intervention with the system makes offline RL a promising paradigm for learning sequential decision-making strategies in many important real-world applications.

Early research in offline RL focused on analyzing approximate value and policy iteration algorithms and had significant overlap with the approximate dynamic programming literature (Munos, 2003, 2007; Munos and Szepesvári, 2008; Antos et al., 2008; Farahmand et al., 2010). These algorithms and their guarantees typically require relatively strong assumptions on both the expressivity of the function class and the exploratoriness of the dataset. For example, the analyses of Fitted Q-Iteration (Ernst et al., 2005; Antos et al., 2007) require the function class to be *closed* under Bellman updates (also known as Bellman-completeness), and the offline data distribution to provide coverage (in some technical sense) over *all* candidate policies (Chen and Jiang, 2019). The former requirement is *non-monotone* in the function class, which shatters the standard machine-learning intuition that a richer function class should always have better (or at least no worse) approximation power; it is also closely related to the instability of RL training and the infamous "deadly triad" (Sutton and Barto, 2018; Wang et al., 2021a). The latter requirement is very likely violated in practice since we have no control over how the historical data is collected (Fujimoto et al., 2019).

Given these considerations, it is desirable to come up with novel algorithms and/or analyses to relax these assumptions. In particular, the ideal assumption on the function class is *realizability*, that there is a target function of interest (such as the optimal value function) and we only require the function class to (approximately) capture such a function. The ideal assumption on the data distribution is *single-policy* coverage, that it is ok for the data to not cover all policies, as long as an optimal (or sufficiently good) policy is covered.

In recent years, significant progress has been made towards providing provable guarantees in offline RL under these relaxed assumptions. In particular, the principle of pessimism in face of uncertainty proves to be useful in designing algorithms that work under single-policy coverage (Liu et al., 2020; Jin et al., 2021; Xie et al., 2021a; Yin et al., 2021; Rashidinejad et al., 2021), but most of the existing pessimistic algorithms require Bellman completeness on the function class. On the other hand, relaxing Bellman completeness to realizability has been difficult: there is merely one existing result that requires only the realizability of optimal value function (Xie and Jiang, 2021), yet their data

*Accepted for the 38th Conference on Uncertainty in Artificial Intelligence* (UAI 2022).

assumption is even stronger than all-policy coverage. In fact, a recent information-theoretic lower bound by Foster et al. (2021) confirms that even with a strong notion of all-policy coverage called (all-policy) concentrability, plus the realizability of value functions for *all* policies, offline RL is still fundamentally intractable.

Despite the lower bound, not all hope is lost. A promising way of breaking the lower bound is to assume the realizability of other functions beyond value functions. Indeed, positive results that are analogues to what we want are established in off-policy evaluation (OPE)—where the goal is to estimate the performance of a target policy from offline data—when additional realizability of *density-ratio* functions is assumed. In particular, Liu et al. (2018); Uehara et al. (2020) show that, as long as the data covers the target policy, and we are given function classes that can represent *both* the value function and the density-ratio function (or marginalized importance weights) of the target policy, it is possible to estimate the performance of the target policy in a sample-efficient manner. One way of using such results for policy learning is to use OPE as a subroutine and optimize a policy using OPE's assessment of the policy's performance. Unfortunately, such a direct application introduces prohibitive expressivity assumptions we wanted to avoid beginning with, such as the realizability of value functions for *all* candidate policies (Jiang and Huang, 2020).

In this paper, we provide sample-efficiency guarantees for offline RL under the desired assumptions, that data is only guaranteed to cover the optimal policy, and the function classes only represent the optimal value function and density ratio, respectively. Our algorithms have a simple procedure that combines marginalized importance sampling (MIS) with pessimism in a novel fashion. The key enabler of our guarantees is an additional *gap* assumption, that there is a nontrivial gap between the values of the (unique) greedy action and the second-best for every state. Similar gap assumptions are common in RL theory to characterize easy problems in which stronger-than-usual guarantees can be obtained. They are often used to achieve logarithmic or constant regret in bandits and tabular online RL (Bubeck and Cesa-Bianchi, 2012; Ok et al., 2018; Slivkins et al., 2019; Lattimore and Szepesvári, 2020; He et al., 2021; Papini et al., 2021), and similar guarantees in offline RL under Bellman-completeness and additional structural assumptions on the value-function class (Hu et al., 2021). They are also used in online RL with function approximation to block exponential error amplification (Du et al., 2019). To our knowledge, our work is the first one to identify the utility of gap assumptions in offline RL with weak function approximation and offer interesting insights into novel aspects and mechanisms of gaps (see Section 5).

**Paper Organization** The rest of the paper is organized as follows: Section 2 introduces preliminary concepts and the problem setting. Section 3 describes the algorithm. Section 4 provides the core analysis. We further extend our results to the setting where the function classes are misspecified (Section 5), and when the gap parameter is unknown but we have access to a small amount of online interactions (Section 6). We conclude the paper with further discussions in Section 7, including a detailed comparison to the concurrent work of Zhan et al. (2022) on the same problem.

## 2 PRELIMINARIES

**Markov Decision Processes (MDPs)** We consider finite horizon episodic MDPs defined in the form of $\mathcal{M} = (\mathcal{X}, \mathcal{A}, P, R, H, x_0)$, where $\mathcal{X} = \mathcal{X}_0 \bigcup \ldots \bigcup \mathcal{X}_{H-1}$ is the layered state space with $\mathcal{X}_h$ denoting the state space at timestep $h$, $\mathcal{A}$ is the action space, $P = (P_0, \ldots, P_{H-1})$ is the transition function with $P_h : \mathcal{X}_h \times \mathcal{A} \to \Delta(\mathcal{X}_{h+1})$, $R = (R_0, \ldots, R_{H-1})$ is the reward function with $R_h : \mathcal{X}_h \times \mathcal{A} \to [0, 1]$, $H$ is the length of horizon, and $x_0$ is the fixed initial distribution.[1] We assume the state and action spaces are finite but can be arbitrarily large, and $\Delta(\cdot)$ denotes the probability simplex over a finite set. We define a policy $\pi = \{\pi_0, \ldots, \pi_{H-1}\}$, where for each $h \in [H]$, $\pi_h : \mathcal{X}_h \to \Delta(\mathcal{A})$ is the policy at timestep $h$ and we use $[H]$ to denote $\{0, \ldots, H-1\}$. With a slight abuse of notation, when $\pi_h(\cdot)$ is a deterministic policy, we assume $\pi_h(\cdot) : \mathcal{X}_h \to \mathcal{A}$. Policy $\pi$ induces a distribution over trajectories from the initial state distribution, which we denote as $\mathrm{Pr}_\pi(\cdot)$ and can be described as starting with $x_0$ and $a_h \sim \pi(\cdot|x_h), r_h = R_h(x_h, a_h), x_{h+1} \sim P_h(\cdot|x_h, a_h), \forall h \in [H]$. As a convention, we will use $x_h, a_h, r_h$ to refer the state, action, and reward at timestep $h$ (thus $x_h \in \mathcal{X}_h$). The performance of a policy is measured by its expected return, defined as $v^\pi := \mathbb{E}_\pi[\sum_{h=0}^{H-1} r_h]$, where the expectation is taken with respect to $\mathrm{Pr}_\pi(\cdot)$. For any $f_h \in \mathbb{R}^{\mathcal{X}_h \times \mathcal{A}}$, we use $\pi_{f_h}(x_h) := \mathrm{argmax}_{a_h \in \mathcal{A}} f_h(x_h, a_h)$ to denote its greedy policy at timestep $h$. Among all policies, there always exists a policy, denoted as $\pi^*$, that maximizes the return from all starting states simultaneously. This policy is the greedy policy of the optimal action-value (or Q-) function, $Q^* = (Q_0^*, \ldots, Q_{H-1}^*)$, i.e., $\pi^* = \pi_{Q^*} := (\pi_{Q_0^*}, \ldots, \pi_{Q_{H-1}^*})$. $Q^*$ is the unique solution to the Bellman optimality equations $Q_h^* = \mathcal{T}_h Q_{h+1}^*$, where $\mathcal{T}_h : \mathbb{R}^{\mathcal{X}_{h+1} \times \mathcal{A}} \to \mathbb{R}^{\mathcal{X}_h \times \mathcal{A}}$ is the Bellman optimality operator: $\forall f_{h+1} \in \mathbb{R}^{\mathcal{X}_{h+1} \times \mathcal{A}}, (\mathcal{T}_h f_{h+1})(x_h, a_h) := R_h(x_h, a_h) + \mathbb{E}_{x_{h+1} \sim P_h(\cdot|x_h, a_h)}[\max_{a_{h+1}} f_{h+1}(x_{h+1}, a_{h+1})]$. We can similarly define policy-specific Q-functions $Q^\pi$ and their state-value function counterparts, namely $V^*$ and $V^\pi$. Another useful concept is the notion of state-action occupancy of a policy $\pi$, $d_h^\pi(x_h', a_h') := \mathrm{Pr}_\pi(x_h = x_h', a_h = a_h')$. As a shorthand, we define $d_h^* := d_h^{\pi^*}$ and use $a_{i:j}$ to refer actions $a_i, \ldots, a_j$.

---

[1] We consider fixed initial state and deterministic reward function. They can be easily generalized to the stochastic case.

**Offline RL** We consider a standard theoretical setup for offline RL, where we are given a dataset $\mathcal{D} = \mathcal{D}_0 \bigcup \ldots \bigcup \mathcal{D}_{H-1}$ with the form $\mathcal{D}_h = \{x_h^{(i)}, a_h^{(i)}, r_h^{(i)}, x_{h+1}^{(i)}\}_{i=1}^n$ and $\mathcal{D}_h$ consists of $\{x_h, a_h, r_h, x_{h+1}\}$ tuples sampled i.i.d. from the following generative process: $(x_h, a_h) \sim d_h^D, r_h = R_h(x_h, a_h), x_{h+1} \sim P_h(\cdot|x_h, a_h)$. Note that $r_h$ and $x_{h+1}$ are generated according to the MDP reward and transition functions, and $d_h^D$ fully determines the quality and coverage of the data distribution. For a given policy $\pi$, $w_h^\pi(x_h, a_h) := d_h^\pi(x_h, a_h)/d_h^D(x_h, a_h)$ measures how well $d_h^D$ covers the occupancy induced by $\pi$ at timestep $h$ and is often known as the density-ratio function or the marginalized importance weight. It plays an important role in offline RL algorithms and analyses. As another shorthand, we use notation $w^\pi = (w_0^\pi, \ldots, w_h^\pi)$ to denote the density-ratio function over all timesteps and notation $w^* := w^{\pi^*}$ to denote the density ratio of the optimal policy.

**Function Approximation** We consider the function approximation setting, where we are given a function class $\mathcal{F} = \mathcal{F}_0 \times \ldots \times \mathcal{F}_{H-1}$ with $\mathcal{F}_h \subseteq (\mathcal{X}_h \times \mathcal{A} \to \mathbb{R}), \forall h \in [H]$ and a weight function class $\mathcal{W} = \mathcal{W}_0 \times \ldots \times \mathcal{W}_{H-1}$ with $\mathcal{W}_h \subseteq (\mathcal{X}_h \times \mathcal{A} \to \mathbb{R}), \forall h \in [H]$. We assume these are finite classes and use $\log(|\mathcal{F}|)$ and $\log(|\mathcal{W}|)$ to measure their statistical capacities. The extension to continuous or infinite classes with a covering argument is standard. By default, for any $f \in \mathcal{F}$, we assume $f_H = \mathbf{0}$ for technical simplicity and use $V_f$ to denote its induces state-value function, i.e., $V_f(x_h) = \max_{a_h \in \mathcal{A}} f_h(x_h, a_h)$. We will also use $\pi_f(x_h)$ instead of $\pi_{f_h}(x_h)$ for simplicity since only $f_h$ operates on $x_h \in \mathcal{X}_h$ and there is no confusion.

## 3 ALGORITHM

In this section, we introduce our algorithm PABC (Pessimism under Average Bellman error Constraints), whose pseudo-code is given in Algorithm 1. The algorithm takes two steps: a prescreening step (line 1), followed by the main step (line 2). We first give an intuition for the main step, deferring the explanation of the prescreening step and the related gap definitions to the later part of this section.

The main step (line 2) runs a constrained optimization to select a function $\hat{f} \in \mathcal{F}$, whose greedy policy is the output. The objective of the optimization minimizes the value at the initial state, which is a form of initial-state pessimism (Xie et al., 2021a; Zanette et al., 2021) and proved to be useful in handling insufficient data coverage. The constraints eliminate functions with large *average Bellman errors* (Jiang et al., 2017; Xie and Jiang, 2020).

**Average Bellman Error Constraints** To provide intuition, we know that $Q^*$ has 0 average Bellman errors for all state-action pairs, that is, $\forall h \in [H], x_h \in \mathcal{X}_h, a_h \in \mathcal{A}$, $(Q_h^* - \mathcal{T}_h Q_{h+1}^*)(x_h, a_h) = 0$. Thus it also has 0 average

---

**Algorithm 1** PABC (Pessimism under Average Bellman error Constraints)

**Input:** threshold $\alpha > 0$, gap parameter $C_{\text{gap}}$, function class $\mathcal{F}$, weight function class $\mathcal{W}$, and dataset $\mathcal{D}$.

1: Perform prescreening according to input $C_{\text{gap}}$:

$$\mathcal{F}(C_{\text{gap}}) := \{f \in \mathcal{F} : \text{gap}(f) \geq C_{\text{gap}}\}. \quad (1)$$

2: Find the pessimism value function in $\mathcal{F}(C_{\text{gap}})$ subject to average Bellman error constraints

$$\hat{f} = \underset{f \in \mathcal{F}(C_{\text{gap}})}{\arg\min} f_0(x_0, \pi_f(x_0))$$

$$\text{s.t.} \max_{w \in \mathcal{W}, h \in [H]} |\mathcal{L}_\mathcal{D}(f, w, h)| \leq \alpha, \quad (2)$$

where the empirical loss $\mathcal{L}_\mathcal{D}(f, w, h)$ is defined as

$$\mathcal{L}_\mathcal{D}(f, w, h) = \frac{1}{n} \sum_{i=1}^n [w_h(x_h^{(i)}, a_h^{(i)})(f_h(x_h^{(i)}, a_h^{(i)})$$
$$- r_h^{(i)} - f_{h+1}(x_{h+1}^{(i)}, \pi_f(x_{h+1}^{(i)})))]. \quad (3)$$

**Output:** policy $\pi_{\hat{f}}$ and return estimation $\hat{f}_0(x_0, \pi_{\hat{f}}(x_0))$.

---

Bellman errors under any distribution $\nu_h$ at timestep $h$:

$$\mathbb{E}_{(x_h, a_h) \sim \nu_h}[(Q_h^* - \mathcal{T}_h Q_{h+1}^*)(x_h, a_h)] = 0.$$

This holds even if $\nu_h$ is an unnormalized distribution. Therefore, we can safely eliminate any candidate function $f \in \mathcal{F}$, if it has a large average Bellman error $\mathbb{E}_{\nu_h}[f_h - \mathcal{T}_h f_{h+1}]$ under any (possibly unnormalized) distribution $\nu_h$. Unlike the more standard versions of Bellman errors such as $\mathbb{E}_{\nu_h}[(f_h - \mathcal{T}_h f_{h+1})^2]$, which squares the Bellman error in each state before taking expectation and cannot be directly estimated due to the infamous *double-sampling* difficulty (Baird, 1995; Farahmand and Szepesvári, 2011), the average Bellman error can be easily estimated. In the algorithm, we consider a variety of (possibly unnormalized) distributions $\nu_h = w_h \cdot d_h^D$ for $w \in \mathcal{W}, h \in [H]$. Since the average Bellman error can only be empirically approximated (see Eq. (3)), we relax the constraints and allow a threshold $\alpha$ to incorporate the statistical errors. We note that the constraint alone is similar to the MABO algorithm by Xie and Jiang (2020), but they do not use pessimism and cannot handle insufficient data coverage. They also assume that $\mathcal{W}$ is sufficiently rich to approximate $w^{\pi_f}$ *for all* $f \in \mathcal{F}$, and a main goal of our work is to avoid such "for all" assumptions.

**Gap and Prescreening** As mentioned in the introduction, a key assumption that enables our results is a gap assumption on value functions. To prepare for the discussion, we define the gap of a function as follows:

**Definition 1** (Gap)**.** *For any* $f = (f_0, \ldots, f_{H-1})$, *we define its gap at timestep* $h \in [H]$ *and state* $x_h \in \mathcal{X}_h$ *as follows: If* $\arg\max_{a_h \in \mathcal{A}} f_h(x_h, a_h)$ *is unique, then we define* $\mathrm{gap}(f; h, x_h) := \min_{a_h \neq \pi_f(x_h)} f_h(x_h, \pi_f(x_h)) - f(x_h, a_h)$. *Otherwise, we define* $\mathrm{gap}(f; h, x_h) := 0$.

*The gap of* $f$ *is then defined as*

$$\mathrm{gap}(f) := \min_{h \in [H], x_h \in \mathcal{X}_h} \mathrm{gap}(f; h, x_h).$$

As we see, this definition of the gap is similar to the one used in prior works (Simchowitz and Jamieson, 2019; Mou et al., 2020; Du et al., 2019; He et al., 2021; Yang et al., 2021; Hu et al., 2021; Wang et al., 2021b; Papini et al., 2021; Wu et al., 2021), except that we require a unique optimal action for the gap to be non-zero. A motivating example of similar gap assumptions in other areas of RL theory can be found in Wu et al. (2021). With Definition 1, we can now define the minimum gap of a function class:

**Definition 2** (Gap of a function class)**.** *Given a function class* $\mathcal{G} = \mathcal{G}_0 \times \ldots \times \mathcal{G}_{H-1}$, *where* $\mathcal{G}_h \subseteq (\mathcal{X}_h \times \mathcal{A} \to \mathbb{R}), \forall h \in [H]$, *we define its gap as*

$$\mathrm{gap}(\mathcal{G}) := \min_{g \in \mathcal{G}} \mathrm{gap}(g).$$

Prior theoretical results relying on similar gap assumptions often make such assumptions on the true optimal value function $Q^*$ (Simchowitz and Jamieson, 2019; Yang et al., 2021). As we will see in our analyses, however, what is really important for us is that the *learned* function $\hat{f}$ has a large gap, not the true $Q^*$. Since we have no control over which $f$ in the function class will be finally chosen, we perform the pre-screening step in line 1 to eliminate functions with the gap lower than a pre-defined threshold $C_{\mathrm{gap}} \geq 0$. It is immediate to see that $\mathrm{gap}(\mathcal{F}(C_{\mathrm{gap}})) \geq C_{\mathrm{gap}}$. Of course, this runs into the risk of eliminating $Q^*$, and if we do not want any misspecification, we need to ensure $Q^* \in \mathcal{F}(C_{\mathrm{gap}})$, which requires that $C_{\mathrm{gap}} \leq \mathrm{gap}(Q^*)$. For clarity, in Section 4.3 we will assume that we have the knowledge of $\mathrm{gap}(Q^*)$ and can set $C_{\mathrm{gap}}$ accordingly, while later in Section 6 we show how to handle unknown $\mathrm{gap}(Q^*)$. Moreover, as we will see in Section 5, when we allow misspecification errors in the analysis, $\mathrm{gap}(Q^*)$ and $C_{\mathrm{gap}}$ become disentangled, which leads to some interesting implications.

## 4 MAIN GUARANTEES

In this section, we present the main sample complexity results of our algorithms. We start with a weak version of guarantee by showing that our algorithm can identify $v^*$, the optimal expected return at the initial state, with polynomial samples under realizability and single-policy coverage assumptions, even without any gap assumption (Section 4.1). Such a result will also be useful when we

handle the unknown gap setting later in Section 6. Then, Section 4.2 provides an algorithm-specific counterexample to show that our algorithm fails to find a near-optimal policy under these assumptions, motivating the necessity of the gap assumption. Finally, Section 4.3 provides the main result of this paper under the additional gap assumption.

### 4.1 ESTIMATING THE OPTIMAL EXPECTED RETURN

We first show how to identify $v^*$, the optimal expected return of the problem, *without* needing the gap assumption. In this case, we will run Algorithm 1 with $C_{\mathrm{gap}} = 0$, that is, without the prescreening step. To our knowledge, there is no prior work that can achieve this goal under our weak assumptions.[2] Despite not producing a near-optimal policy, this procedure and guarantee allows us to check whether any given policy is close to optimal, assuming we can evaluate the policy's performance by off-policy evaluation or a small amount of online interactions. This capability can be very useful especially in certain model selection scenarios (see e.g., Modi et al., 2020, Section 5). Indeed, we will reuse this result later in Section 6 to handle the unknown gap setting.

We start by introducing the assumptions. The first two are the standard realizability assumptions.

**Assumption 1** (Realizability of $\mathcal{F}$)**.** *We assume* $Q^* = (Q_0^*, \ldots, Q_{H-1}^*) \in \mathcal{F}$.

**Assumption 2** (Realizability of $\mathcal{W}$)**.** *We assume* $w^* = (w_0^*, \ldots, w_{H-1}^*) \in \mathcal{W}$.

We make these assumptions exact for now to allow for a clean presentation of the main results and core proof ideas, and defer the handling of misspecification errors to Section 5. Also, following the arguments in Uehara et al. (2020); Xie and Jiang (2020), Assumption 2 can be further relaxed such that $w^*$ only needs to lie in the convex hull of $\mathcal{W}$.

Next, we introduce the standard boundedness assumptions.

**Assumption 3** (Boundness of $\mathcal{F}$)**.** *For any* $f \in \mathcal{F}$, *we assume* $f_h \in (\mathcal{X}_h \times \mathcal{A} \to [0, H - h]), \forall h \in [H]$.

**Assumption 4** (Boundness of $\mathcal{W}$)**.** *For any* $w \in \mathcal{W}$, *we assume* $\|w_h\|_\infty \leq C, \forall h \in [H]$.

Assumption 2 and Assumption 4 together immediately imply that our data covers $\pi^*$:

$$\frac{d_h^*(x_h, a_h)}{d_h^D(x_h, a_h)} \leq C, \forall h \in [H], x_h \in \mathcal{X}_h, a_h \in \mathcal{A}.$$

This version of coverage is often called $\pi^*$-concentrability (Scherrer, 2014; Xie et al., 2021b; Rashidinejad et al., 2021;

---

[2]We note that under Zhan et al. (2022)'s assumptions, their algorithm, with regularization removed, can also identify $v^*$.

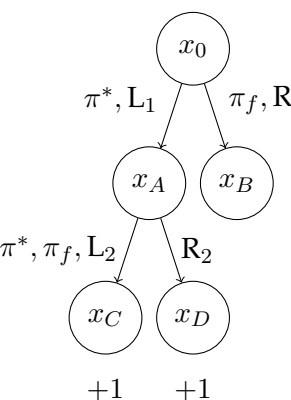

Figure 1: Algorithm-specific counterexample without the gap assumption.

Zhan et al., 2022). As we will see when we consider misspecification errors in Section 5, we do not really need our data to satisfy $\pi^*$-concentrability, and the definition of coverage can be relaxed using the structure and generalization effects of $\mathcal{F}$ similarly to Jin et al. (2021); Xie et al. (2021b).

With all the above assumptions, we are ready to state our first result formally below. The proof is deferred to Appendix A.2.

**Theorem 1** (Sample complexity of estimating $v^*$). *Suppose Assumptions 1, 2, 3, 4 hold and the total samples $nH$ satisfies*

$$nH \geq \frac{8C^2H^5\log(2|\mathcal{F}||\mathcal{W}|H/\delta)}{\varepsilon^2}.$$

*Then with probability at least $\geq 1 - \delta$, running Algorithm 1 with $C_{\text{gap}} = 0$ and $\alpha = \varepsilon/(2H)$ guarantees*

$$|V_{\hat{f}}(x_0) - v^*| \leq \varepsilon.$$

### 4.2 ALGORITHM-SPECIFIC COUNTEREXAMPLE

Despite being able to identify $v^*$, we show that Algorithm 1 cannot be guaranteed to learn a near-optimal policy without further assumptions, even with infinite data. As we will see, a key aspect of the construction is a tie between the values of different actions, so such counterexamples can be effectively excluded by assuming a unique optimal action.

The counterexample is given in Figure 1. Circles denote states and arrows denote actions with deterministic transitions, and states without arrows have a default null action. There are only $+1$ rewards at states $x_C$ and $x_D$, while the rewards are 0 everywhere else. Taking $L_1$ at $x_0$ deterministically transits to $x_A$ and we omit the remaining specifications as they are clearly indicated in the figure.

It is easy to see that the optimal policy $\pi^*$ takes action $L_1$ at state $x_0$. By adversarial tie breaking, we assume $\pi^*$ takes

action $L_2$ at state $x_A$. We construct a bad function $f$, which only differs from $Q^*$ at $(x_0, R_1)$ and $(x_B, \text{null})$ by setting $f_0(x_0, R_1) = 1$ and $f_1(x_B, \text{null}) = 1$. By adversarial tie breaking, we assume $\pi_f(x_0) = R_1$ and $\pi_f(x_A) = L_2$. We immediately have a realizable class $\mathcal{F} = \{Q^*, f\}$. It is easy to verify that $\pi_f$ is not $\varepsilon$-optimal for any $\varepsilon < 1$ because it deviates from the optimal branch at $x_0$. In addition, we let data $d^D$ covers $(x_0, L_1)$, $(x_A, L_2)$, $(x_C, \text{null})$. For the weight function, we define an *invalid* weight function $w_{\text{bad}}$ that puts all weight on $(x_0, R_1)$, $(x_A, L_2)$, $(x_C, \text{null})$ in each level respectively. Then we also have a realizable class $\mathcal{W} = \{w^*, w_{\text{bad}}\}$.

As $\text{gap}(Q^*) = 0$ in this counterexample, no function will be ruled out in the prescreening step (line 1). Since both $f$ and $Q^*$ have zero population average Bellman error under $\mathcal{W}$, and $f_0(x_0, \pi_f(x_0)) = Q_0^*(x_0, \pi_{Q^*}(x_0)) = 1$, either of them can be the $\hat{f}$ learned in Algorithm 1, but returning $\pi_f$ leads to failure of learning. We note that the reason for failure is that no data covers the state $\pi_f$ visits.

**Additional Consistency Constraints**   In our setting, there are additional constraints that one can add to ensure some form of consistency. For example, for any $f \in \mathcal{F}$, we can additionally require that there exists $w \in \mathcal{W}$ that is consistent with $\pi_f$, since $w^* \in \mathcal{W}$ should only give non-zero weight to actions chosen by $\pi_f$ (i.e., $\forall h \in [H], w_h(x_h, a_h) = 0$ if $a_h \neq \pi_f(x_h)$). In addition, as we can estimate $v^*$ with the assumptions in Theorem 1 and we know $\mathbb{E}_{d^D}[w^* \cdot R] = v^*$, we can eliminate any $w \in \mathcal{W}$ that violates this condition. While these constraints are reasonable (or at least harmless) and may be of independent interest, we can verify that they do not help with this counterexample, which implies our algorithm fails even under these additional consistency checks.

### 4.3 LEARNING A NEAR-OPTIMAL POLICY

As mentioned above, a key aspect of the counterexample is a tie between the values of actions. In this section, we show that a positive gap assumption not only excludes the counterexample, but enables a general guarantee for learning near-optimal policies with our Algorithm 1.

**Assumption 5** (Gap of $Q^*$). *The gap of $Q^*$ satisfies*

$$\text{gap}(Q^*) > 0.$$

Here the implicit assumption is that we want $\text{gap}(Q^*)$ to be sufficiently large, as our later sample complexity guarantees will scale inversely with $\text{gap}(Q^*)$. Note that Assumption 5 is stronger than the standard gap assumption in the literature (Simchowitz and Jamieson, 2019; Yang et al., 2021; Hu et al., 2021). Compared with their definition, we additionally assume the optimal action is unique at each state. On the

other hand, these papers require additional strong assumptions (e.g., linear MDPs, Bellman-completeness, or pointwise convergence) or focus on the tabular setting, whereas we handle general function approximation in offline RL under weak realizability-type assumptions. Plus, the technical mechanisms under which the gap plays a role in the analyses are very different, so the assumptions are not really comparable.

For now, we assume $\mathrm{gap}(Q^*)$ is known and will later handle the case of an unknown gap Section 6. Plus, as a side effect of handling misspecification errors, Section 5 will lift the stringent gap assumption in a novel and interesting manner.

We now state the guarantee of learning a near-optimal policy under the gap assumption. A sketch of proof is provided after the theorem statement, while the complete proof is deferred to Appendix A.3.

**Theorem 2** (Sample complexity of learning a near-optimal policy). *Suppose Assumptions 1, 2, 3, 4, 5 hold and the total number of samples $nH$ satisfies*

$$nH \geq \frac{8C^2 H^7 \log(2|\mathcal{F}||\mathcal{W}|H/\delta)}{\varepsilon^2 \mathrm{gap}(Q^*)^2}.$$

*Then with probability at least $\geq 1 - \delta$, running Algorithm 1 with $\alpha = \varepsilon \mathrm{gap}(Q^*)/(2H^2)$ and $C_{\mathrm{gap}} = \mathrm{gap}(Q^*)$ guarantees*

$$v^{\pi_{\hat{f}}} \geq v^* - \varepsilon.$$

**Proof sketch of Theorem 2** As standard, all our results depend on a high-probability concentration event, that $|\mathcal{L}_\mathcal{D}(f, w, h) - \mathbb{E}[\mathcal{L}_\mathcal{D}(f, w, h)]| \leq \varepsilon_{\mathrm{stat},n}$ holds for all $f \in \mathcal{F}, w \in \mathcal{W}, h \in [H]$ with high probability; the detailed expression of $\varepsilon_{\mathrm{stat},n}$ is given in Lemma 1. From Lemma 1, for any $f \in \mathcal{F}$ that satisfies all constraints in Algorithm 1, we can guarantee the population loss to be small, that is, $|\mathbb{E}[\mathcal{L}_\mathcal{D}(f, w, h)]| \leq \varepsilon_{\mathrm{stat},n} + \alpha$.

The central step of our proof is to use a telescoping argument and the gap assumption to establish the following inequality:

$$V_0^*(x_0) \geq V_{\hat{f}}(x_0) \geq V_0^*(x_0) - H(\varepsilon_{\mathrm{stat},n} + \alpha)$$
$$+ \mathrm{gap}(Q^*)\mathbb{E}\left[\sum_{h=0}^{H-1} \mathbf{1}\{\pi_{\hat{f}}(x_h) \neq \pi^*(x_h)\} \mid \pi^*\right].$$

This implies the policy deviation can be bounded as

$$\mathbb{E}\left[\sum_{h=0}^{H-1} \mathbf{1}\{\pi_{\hat{f}}(x_h) \neq \pi^*(x_h)\} \mid \pi^*\right] \leq \frac{H(\varepsilon_{\mathrm{stat},n} + \alpha)}{\mathrm{gap}(Q^*)}.$$

The LHS of this inequality is the probability that the learned policy $\pi_{\hat{f}}$ disagrees with the optimal policy $\pi^*$, along the distribution induced by $\pi^*$. From here, we can apply the RL-to-supervised-learning (SL) reduction in imitation learning (e.g., Theorem 2.1 in Ross and Bagnell (2010)) to translate it to the final performance difference bound between $\pi_{\hat{f}}$ and $\pi^*$. We also provide a different proof in Appendix A.3, which itself may be of independent interest.

## 5 ROBUSTNESS TO MISSPECIFICATION

We now consider the case when $Q^*$ and $w^*$ may not exactly belong to $\mathcal{F}$ and $\mathcal{W}$, but can be reasonably approximated up to small errors. More often than not, such robustness results in RL theory are nothing but routine exercises where the proofs are largely straightforward extensions of those for the exact case. In our case, however, the misspecification analyses reveal an interesting phenomenon of disentangling the true gap of $Q^*$ and that of $\mathcal{F}$, and how our gap and coverage assumptions can be relaxed in nontrivial ways.

We start with defining the approximation errors of our function classes. Inspired by Xie and Jiang (2020), we define the approximation error of $\mathcal{W}$ as

$$\varepsilon_\mathcal{W} = \min_{w \in \mathcal{W}} \max_{f \in \mathcal{F}} \max_{h \in [H]} |\mathbb{E}_{d_h^D}[w_h \cdot (f_h - \mathcal{T}_h f_{h+1})]$$
$$- \mathbb{E}_{d_h^*}[f_h - \mathcal{T}_h f_{h+1}]| \qquad (4)$$

and use $\tilde{w}^*$ to denote the best approximator in $\mathcal{W}$ that obtains the minimum. The expression inside the min-max-max measures the difference between $d_h^D \cdot w_h$ and $d_h^*$, using $f_h - \mathcal{T}_h f_{h+1}$ for $f \in \mathcal{F}$ as discriminators. When $w^* \in \mathcal{W}$ (Assumption 2), $d_h^D \cdot w_h^* = d_h^*$ (because $w_h^*$ is defined as $d_h^*/d_h^D$), so we have $\varepsilon_\mathcal{W} = 0$. However, the opposite direction is *not* always true: it is entirely possible to achieve $\varepsilon_\mathcal{W} = 0$ when $w^* \notin \mathcal{W}$ (or even when $d_h^*(x_h, a_h)/d_h^D(x_h, a_h) = \infty$ for some $(x_h, a_h)$ and $w^*$ does not exist), as long as $d_h^D \cdot \tilde{w}_h^*$ and $d_h^*$ can not be distinguished by $f_h - \mathcal{T}_h f_{h+1}$ for $f \in \mathcal{F}$ as *discriminators*.[3] We also provide an example in Appendix E. Note that since our data coverage assumption is implicitly made through the realizability and boundedness of $\mathcal{W}$ (see the discussion below Assumption 4), this means that our data coverage assumption is also relaxed using the information of $\mathcal{F}$, which is a common characteristics of recent results in offline RL (e.g., Xie et al. (2021a) also use the Bellman error class induced by the value function class as discriminators, which is similar to our definition at a high level), but not enjoyed by the concurrent work of Zhan et al. (2022).

For function class $\mathcal{F}$, we define the approximation error in a way that uses $\mathcal{W}$ as discriminators, plus a term that measures the difference under the initial state $x_0$:

$$\varepsilon_\mathcal{F} = \min_{f \in \mathcal{F}} \max_{w \in \mathcal{W}} \max_{h \in [H]} (|\mathbb{E}_{d_h^D}[w_h \cdot (f_h - \mathcal{T}_h f_{h+1})]|$$
$$+ |f_0(x_0, \pi_f(x_0)) - Q_0^*(x_0, \pi^*(x_0))|) \qquad (5)$$

and use $\tilde{Q}_\mathcal{F}^*$ to denote the best approximator that achieves the minimum value. Under mild regularity assumptions on $\mathcal{W}$,[4] it is straightforward to show that $\varepsilon_\mathcal{F}$ is weaker than $\ell_\infty$

---

[3] The idea of using discriminators has also been explored in Farahmand et al. (2017); Sun et al. (2019); Modi et al. (2020, 2021), but the application is different here.

[4] Namely, $w \geq 0$ and $\mathbb{E}_{d^D}[w] = 1$. The former is trivial and the latter can be easily verified approximately on data.

error up to multiplicative constants:

$$\varepsilon_{\mathcal{F}} \leq 3 \min_{f \in \mathcal{F}} \max_{h \in [H]} \|f_h - Q_h^*\|_\infty, \qquad (6)$$

and a more detailed discussion can be found in Lemma 4. Similarly, we can define the function class $\mathcal{F}(C_{\mathrm{gap}})$ related approximation error $\varepsilon_{\mathcal{F}(C_{\mathrm{gap}})}$ and its best approximator $\tilde{Q}^*_{\mathcal{F}(C_{\mathrm{gap}})}$ by replacing $\mathcal{F}$ with $\mathcal{F}(C_{\mathrm{gap}})$ in Eq. (5).

## 5.1 ESTIMATING THE OPTIMAL EXPECTED RETURN

With the approximation error defined above, we now extend Theorem 1 to the approximate case. Assuming the approximation error $\varepsilon_{\mathcal{F}}$ (or a reasonably tight upper bound of it) is known, we can relax the constraint to ensure that $\tilde{Q}^*_{\mathcal{F}(C_{\mathrm{gap}})}$ is not eliminated and obtain the sample complexity guarantee as in Theorem 3. The full proof is deferred to Appendix B.2. As before, we do not need the gap assumption to identify $v^*$ approximately and can run the algorithm with $C_{\mathrm{gap}} = 0$.

**Theorem 3** (Robust version of Theorem 1). *Suppose Assumptions 3, 4 hold and the total number of samples $nH$ satisfies*

$$nH \geq \frac{8C^2 H^5 \log(2|\mathcal{F}||\mathcal{W}|H/\delta)}{\varepsilon^2}.$$

*Then with probability $1 - \delta$, running Algorithm 1 with $\alpha = \varepsilon/(2H) + \varepsilon_{\mathcal{F}}$ and $C_{\mathrm{gap}} = 0$ guarantees*

$$|V_{\hat{f}}(x_0) - v^*| \leq \varepsilon + H\varepsilon_{\mathcal{F}} + H\varepsilon_{\mathcal{W}}.$$

While we need the knowledge of $\varepsilon_{\mathcal{F}}$ to set $\alpha$, we do not need to know $\varepsilon_{\mathcal{W}}$, which shows a difference between the behaviors of $\mathcal{F}$ and $\mathcal{W}$. This is also the case in the next subsection where we try to learn a near-optimal policy.

## 5.2 LEARNING A NEAR-OPTIMAL POLICY

Similarly, we can also extend Theorem 2 to the misspecified case. Our guarantee is established with a user-specified $C_{\mathrm{gap}}$ parameter and the approximation error related to prescreened class $\mathcal{F}(C_{\mathrm{gap}})$. We provide the sample complexity guarantee in Theorem 4 and the complete proof in Appendix B.3.

**Theorem 4** (Robust version of Theorem 2). *Suppose Assumptions 3, 4 hold and the total number of samples $nH$ satisfies*

$$nH \geq \frac{8C^2 H^7 \log(2|\mathcal{F}||\mathcal{W}|H/\delta)}{\varepsilon^2 C_{\mathrm{gap}}^2}.$$

*Then with probability $1 - \delta$, running Algorithm 1 with a user-specified $C_{\mathrm{gap}}$ and $\alpha = \varepsilon C_{\mathrm{gap}}/(2H^2) + \varepsilon_{\mathcal{F}(C_{\mathrm{gap}})}$ guarantees*

$$v^{\pi_{\hat{f}}} \geq v^* - \varepsilon - \frac{(H^2 + H)\varepsilon_{\mathcal{F}(C_{\mathrm{gap}})} + H^2\varepsilon_{\mathcal{W}}}{C_{\mathrm{gap}}}.$$

Theorem 4 gives us a convenient way to set the gap parameter $C_{\mathrm{gap}}$. We also provide a sample complexity guarantee (Corollary 5) in Section B.4 for the case that $\mathrm{gap}(Q^*)$ and the $\ell_\infty$ approximation error of $\mathcal{F}$ are known.

**Unknown Approximation Errors** Notice that in the robustness results (Theorem 3 and Theorem 4) we require the knowledge of approximation errors $\varepsilon_{\mathcal{F}}$ or $\varepsilon_{\mathcal{F}(C_{\mathrm{gap}})}$ to set the threshold $\alpha$ in PABC (Algorithm 1). In Appendix D we show a variant of PABC based on Lagrangians (PABC-L; Algorithm 1) does not require such knowledge, and still enjoys the same sample complexity guarantees. In PABC-L, the original constraints in Eq. (2) are moved to the objective, thus the threshold $\alpha$ is no longer needed as the input. We refer the reader to Appendix D for the formal description of PABC-L and its results and proofs.

**Relaxed Gap Assumption** An outstanding characteristic of Theorem 4 is that it no longer depends on $\mathrm{gap}(Q^*)$ explicitly, and only depends on $C_{\mathrm{gap}}$, a parameter of our choice. Therefore, it may seem to have lifted Assumption 5 that $\mathrm{gap}(Q^*) > 0$, as we can choose $C_{\mathrm{gap}}$ to be sufficiently large. However, below we show that this issue is more complicated than it may seem, and while our result does relax Assumption 5 in significant ways, it does so in a very nuanced manner.

First of all, in the worst-case scenario, Assumption 5 is still needed to provide non-vacuous guarantees. This is because, if $Q^*$ has no gap, yet we artificially create a large $C_{\mathrm{gap}}$ in our prescreened function class $\mathcal{F}(C_{\mathrm{gap}})$, we could eliminate all the good approximations of $Q^*$. Among the remaining functions, the best $\ell_\infty$ approximation of $Q^*$ must have an $\ell_\infty$ error no less than $C_{\mathrm{gap}}$, and if we plug that into the $\varepsilon_{\mathcal{F}(C_{\mathrm{gap}})}$ term in the approximation guarantee, the $C_{\mathrm{gap}}$ on the numerator and the denominator will cancel out, leaving a constant suboptimality gap which makes the guarantee vacuous.

Having said that, the nuance here is that we do *not* use the most stringent $\ell_\infty$ norm to define $\varepsilon_{\mathcal{F}(C_{\mathrm{gap}})}$, but rather use an average notion of error (Eq. (5)), which is possibly much smaller than the $\ell_\infty$ error (Eq. (6)). Therefore, there are still cases where $\mathrm{gap}(Q^*) = 0$ yet our result yields nontrivial guarantees. As a concrete example, imagine a $Q^*$ that has large gaps in most states, but the gap is $0$ in a few "bad" states. In this case, $\mathrm{gap}(Q^*)$ is $0$. However, there can still exist $\tilde{Q}^*_{\mathcal{F}(C_{\mathrm{gap}})}$ that approximates $Q^*$ well everywhere except on those bad states, and as long as no $w \in \mathcal{W}$ puts significant probabilities on the bad states, we

have $\varepsilon_{\mathcal{F}(C_{\text{gap}})} \ll C_{\text{gap}}$ and hence Theorem 4 will provide meaningful guarantees.

# 6 HANDLING THE UNKNOWN GAP PARAMETER WITH ONLINE ACCESS

In this section, we extend the main algorithm and analyses in Section 4 in a different direction than Section 5. In particular, we are concerned about the fact that Theorem 2 assumes the knowledge of $\text{gap}(Q^*)$. While it is common for offline RL algorithms to have hyperparameters that need to be tuned separately (and this is particularly the case for version-space-based algorithms (Jiang et al., 2017; Xie et al., 2021a)), here we show that we can address the unknown $\text{gap}(Q^*)$ issue by a small amount of additional online interactions for Monte-Carlo policy evaluation. This is particularly interesting as our result provides an example of how one can use a small amount of online interactions to mitigate limitations in purely offline learning, a practically relevant problem that is also of great interest to the RL theory community (Xie et al., 2021b).

---

**Algorithm 2** PABC-OA (PABC with Online Access)

1: **Input**: function class $\mathcal{F}$, weight function class $\mathcal{W}$, and dataset $\mathcal{D}$ (with size $|\mathcal{D}_h| = n, \forall h \in [H]$).
2: **for** $t = 0, 1, \ldots$ **do**
3:     Set $\text{gap}_t^{\text{guess}} = H/2^t$.
4:     Use $n$ and $\text{gap}_t^{\text{guess}}$ to calculate $\varepsilon_t = \sqrt{8C^2H^6\iota(t)/(n(\text{gap}_t^{\text{guess}})^2)}$, where $\iota(t) = \log(24|\mathcal{F}||\mathcal{W}|H \cdot 2^t/\delta)$.
5:     Run Algorithm 1 with $\alpha = \varepsilon_t/(2H)$ and get scalar estimation $\hat{v}_t^*$.
6:     Run Algorithm 1 with $\alpha = \varepsilon_t\text{gap}_t^{\text{guess}}/(2H^2)$ and $C_{\text{gap}} = \text{gap}_t^{\text{guess}}$, and get policy $\hat{\pi}_t$.
7:     Estimate $v^{\hat{\pi}_t}$ by running Monte Carlo algorithm with $\tilde{O}(H^3 \log(1/\delta)/\varepsilon_t^2)$ online samples and denote the estimate as $\hat{v}^{\hat{\pi}_t}$.
8:     **if** $\hat{v}^{\hat{\pi}_t} \geq \hat{v}_t^* - 3\varepsilon_t$ **then**
9:         Output $\hat{\pi}_t$ and terminate.
10:     **end if**
11: **end for**

---

As shown in Algorithm 2, the algorithm PABC-OA (PABC with Online Access) proceeds iteration by iteration. We start with the maximum possible value of the unknown $\text{gap}(Q^*)$. For simplicity, we choose $H$ here, and alternatively we can also use $\max_{f \in \mathcal{F}} \text{gap}(f)$ which is tighter. In iteration $t$, we use $\text{gap}_t^{\text{guess}} = H/2^t$ as the guess of $\text{gap}(Q^*)$ and calculate the desired $\alpha$ according to Theorem 1 to estimate $v^*$ (line 5), or calculate the desired $\alpha$ and $C_{\text{gap}}$ according to Theorem 2 to find a near-optimal policy (line 6). Finally we conduct Monte-Carlo policy evaluation with online samples (line 7). If the stopping condition (line 8) is satisfied, we are guaranteed to learn a near-optimal policy and can terminate

(line 9). Otherwise, we proceed to the next iteration, shrink our guessed value of $\text{gap}_t^{\text{guess}}$, and continue the routine. We can observe an interesting connection between Theorem 1 and Theorem 2, and identifying $v^*$ is indeed useful.

It can be shown that Algorithm 2 will terminate once the guessed value $\text{gap}_t^{\text{guess}} = H/2^t$ drops below the true value $\text{gap}(Q^*)$, which leads to the sample complexity result in Theorem 5. The formal proof can be found in Appendix C.

**Theorem 5** (Sample complexity of learning a near-optimal policy with unknown $\text{gap}(Q^*)$)**.** *Suppose Assumptions 1, 2, 3, 4, 5 hold but $\text{gap}(Q^*)$ is unknown. Assume we have a dataset $\mathcal{D}$ with size $n$ for each $\mathcal{D}_h$ and additional online access to collect*

$$\tilde{O}\left(\frac{n\log(1/\delta)}{C^2H}\right)$$

*samples. Then with probability at least $1 - \delta$, the output policy $\hat{\pi}$ from Algorithm 2 satisfies*

$$v^{\hat{\pi}} \geq v^* - 5\sqrt{\frac{32C^2H^6\iota(\log(2H/\text{gap}(Q^*)))}{n\text{gap}(Q^*)^2}}, \quad (7)$$

*where $\iota(t) = \log(24|\mathcal{F}||\mathcal{W}|H \cdot 2^t/\delta)$.*

The suboptimality in Eq. (7) has the same order (up to polylog terms) as that of running Algorithm 1 with known $\text{gap}(Q^*)$ in Theorem 2. If we set this value to be $\varepsilon'$, i.e., $\varepsilon' := 5\sqrt{\frac{32C^2H^6\iota(\log(2H/\text{gap}(Q^*)))}{n\text{gap}(Q^*)^2}}$, then the number of required online samples is $\tilde{O}\left(\frac{H^5\log(1/\delta)}{(\varepsilon'\text{gap}(Q^*))^2}\right)$, which does not depend on the complexity of the function classes $\mathcal{F}$ and $\mathcal{W}$.

# 7 DISCUSSION AND CONCLUSION

We conclude the paper with a detailed discussion of how our work compares to the closely related concurrent work of Zhan et al. (2022), which also provides a good summary of our contributions and promising future directions.

The very recent work of Zhan et al. (2022) aims at solving the same problem:[5] offline RL under only single-policy coverage and realizability assumptions. Similar to our counterexample in Section 4.2, they also realize the difficulties in the setting where the optimal weight and value functions are realizable in a straightforward manner. Instead of making a gap assumption like we do, they attack the problem from a different angle by introducing regularization into the Lagrangian of the linear program for MDPs.

Despite that the two approaches have some fundamental differences (which we will elaborate further below), it is

---

[5]Their results are in the discounted setting whereas ours in the finite horizon setting, but this is a superficial difference and translating each of the results into the other setting is not difficult.

still worth comparing the nature of the two results. To this end, our approach has several advantages:

1. Regularization changes the definition of the value function in Zhan et al. (2022). In fact, the function they need to realize does not obey any form of Bellman equations, and probably should not be called value functions anymore. This makes their realizability assumption somewhat difficult to interpret and connect to the existing literature. In contrast, we work with the most standard notion of $Q^*$.

2. Due to regularization, the policy learned by Zhan et al. (2022) is generally suboptimal even with infinite data, so the strength of regularization needs to be carefully controlled for the bias-variance trade-off. As a result, when competing with $\pi^*$, their sample complexity rate is $O(1/\varepsilon^6)$, which is much slower than our $O(1/\varepsilon^2)$.

3. Our coverage assumption can be significantly relaxed using the structure of $\mathcal{F}$; see discussion in Section 5. While this is standard in recent offline RL works based on Bellman-completeness assumptions (Jin et al., 2021; Xie et al., 2021a), Zhan et al. (2022)'s guarantee relies on the boundedness of the raw density ratios and does not enjoy such a relaxation.

That said, Zhan et al. (2022)'s result is also attractive in several aspects:

1. They do not require gap assumptions. While similar gap assumptions are standard in RL theory literature, it is unclear how prevalent it is in real problems and how algorithms that depend on gap assumptions perform in problems when the assumption is violated.

2. Our guarantees only hold if the data covers $\pi^*$ (though the notion of coverage can be relaxed using a structure of $\mathcal{F}$, as mentioned above). In comparison, Zhan et al. (2022) can still provide meaningful guarantees even when $\pi^*$ is not covered by data, in which case they compete with the best policy under data coverage.

3. Regarding computation, their algorithm is a convex-concave minimax optimization problem when the function classes are convex. In comparison, the computational characteristics of our method are less clear, though we note that a Lagrangian form of our main step (line 2) (see Appendix D for details) is similar to the kind of minimax optimization commonly found in the MIS literature (Nachum et al., 2019; Uehara et al., 2020; Yang et al., 2020; Jiang and Huang, 2020).

We reiterate that these comparisons are made only on the results themselves. The two works take fundamentally different approaches and are of independent interests. For example, despite that both works use density-ratio functions, Zhan et al. (2022)'s method is based on the linear programming (LP)-formulation of MDPs where the optimal

state-value function $V^*$ is modeled, whereas we model the optimal Q-function $Q^*$. This difference is more significant than it may seem, as the LP formulation and the Bellman optimality equations for $Q^*$ are very different foundations for designing learning algorithms, and the gap assumption only makes sense for Q-functions and cannot be used in state-value functions. That said, it will be interesting to investigate if the two works can borrow each other's ideas to address their own weaknesses, which we leave to future investigation.

## Acknowledgements

The authors thank Akshay Krishnamurthy for helpful discussions. NJ acknowledges funding support from ARL Cooperative Agreement W911NF-17-2-0196, NSF IIS-2112471, NSF CAREER award, and Adobe Data Science Research Award.

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
