# OpenReview forum: "Offline Reinforcement Learning Under Value and Density-Ratio Realizability: The Power of Gaps"
_auai.org/UAI/2022/Conference — UAI 2022 Poster_

### Official Review · Reviewer_wmQa · 2022-04-12

**Q2(1) Originality/Novelty:** 3
**Q2(2) Significance/Impact:** 3
**Q2(3) Correctness/Technical Quality:** 3
**Q2(6) Clarity Of Writing:** 4
**Q6 Overall Score:** 6
**Q8 Confidence In Your Score:** 1

**Q1 Summary And Contributions:**

- The authors provide a unique solution for sample-efficiency guarantees under certain conditions in theoretical RL
- Provide an algorithm that utilizes a novel gap assumption for the basis of much of their proof-based work

**Q10 Ethical Concerns (Optional):**

None.

**Q2 Assessment Of The Paper:**

More detailed information regarding each of these aspects is given below:

**Q2(5) Reproducibility:**

4: Excellent: Key resources (e.g., proofs, code, data) are available and key details (e.g., proof sketches, experimental setup) are comprehensively described for competent researchers to confidently and easily reproduce the main results.

**Q3 Main Strengths:**

- Strong theoretical guarantees
- Utilize a gap framework which is a common framework employed in theoretical RL for a novel purpose
- Extensive literature review and framing, which is helpful for a non-expert in the space such as myself

**Q4 Main Weakness:**

As a practitioner, the lack of experimental results was difficult, but I understand this is a theory-focussed paper.

**Q5 Detailed Comments To The Authors:**

Solid paper with interesting work.

**Q7 Justification For Your Score:**

I'm not an expert in this space and this is honestly a guess.

**Q9 Complying With Reviewing Instructions:**

1: Yes.

---

### Official Review · Reviewer_yeSY · 2022-04-12

**Q2(1) Originality/Novelty:** 3
**Q2(2) Significance/Impact:** 3
**Q2(3) Correctness/Technical Quality:** 3
**Q2(6) Clarity Of Writing:** 3
**Q6 Overall Score:** 7
**Q8 Confidence In Your Score:** 2

**Q1 Summary And Contributions:**

The authors develop a new offline RL algorithm that provides guarantees under a novel set of assumptions:
* single-policy coverage and realizability (that data is only guaranteed to cover the optimal policy, and the function classes only represent the optimal value function and density ratio, respectively)
* a novel "gap assumption" – that the best and second best have a nontrivial gap in performance. This applies specifically to the policy that is learned, not the true best policy


**Q2 Assessment Of The Paper:**

More detailed information regarding each of these aspects is given below:

**Q2(5) Reproducibility:**

3: Good: Key resources (e.g., proofs, code, data) are available and key details (e.g., proofs, experimental setup) are sufficiently well-described for competent researchers to confidently reproduce the main results.

**Q3 Main Strengths:**

* mathematical as well as intuitive justification
* fair treatment of a related work (Zhan et al 2022)


**Q4 Main Weakness:**

* it would be useful to see an example of an application of the algorithm to an offline RL problem


**Q5 Detailed Comments To The Authors:**

* are there particular real-world (or benchmark) offline RL algorithms where we might expect these assumptions to hold (especially the novel "gap assumption" that you propose)?

**Q7 Justification For Your Score:**

The authors seem to propose both a novel algorithm as well as a novel theoretical framework for thinking about offline RL. However, I am not confident about this score or this judgment, because I was not previously familiar with the topic area.

**Q9 Complying With Reviewing Instructions:**

1: Yes.

---

### Official Review · Reviewer_hSpd · 2022-04-23

**Q2(1) Originality/Novelty:** 3
**Q2(2) Significance/Impact:** 2
**Q2(3) Correctness/Technical Quality:** 3
**Q2(6) Clarity Of Writing:** 3
**Q6 Overall Score:** 5
**Q8 Confidence In Your Score:** 3

**Q1 Summary And Contributions:**

This paper shows the theoretical guarantees of a simple pessimistic algorithm under a gap assumption. The guarantees only requires the coverage of the optimal policy instead of all policies and the realizability of the optimal value and density-ratio functions.

**Q2 Assessment Of The Paper:**

More detailed information regarding each of these aspects is given below:

**Q2(5) Reproducibility:**

3: Good: Key resources (e.g., proofs, code, data) are available and key details (e.g., proofs, experimental setup) are sufficiently well-described for competent researchers to confidently reproduce the main results.

**Q3 Main Strengths:**

This paper works on marginalized importance sampling (MIS), by extending several previous work and achieves great progress. The proof is technically sound as I checked.



**Q4 Main Weakness:**

As shown in an recent experimental submission "StarCraft II Unplugged: Large Scale Offline Reinforcement Learning", MIS algorithm family seems to be mathematically strict, but poor in scalability. This can be alleviated if there exists some medium-scale problems where other off-policy/offline methods can't work, and MIS-based methods can work well. Can the authors propose such application scenarios?

Since an earlier paper Zhan et al. (2022) studies the same problem, can the authors design some examples to compare the results in this paper and Zhan et al. (2022)? To me, the assumptions in this paper are more restrictive than those in Zhan et al. (2022). If so, are there any examples showing that even if the assumptions are not satisfied, the proposed algorithm is still applicable? In addition, besides comparing the (upper) bounds, it would be great to see some experiments that compare the real performances between the proposed algorithms in both papers and other relevant papers.

**Q5 Detailed Comments To The Authors:**

The theoretical merit is obvious. The paper would be strengthened if the application potential can be made clearer.

Some minor comments:
1. It seems that the definition of [H] in this paper is {0,1,...,H-1} instead of {1,2,...,H}. If so, it is necessary to define it.
2. For the sake of general audience, it is better to define MIS (marginalized importance sampling), LP (linear programming), etc before using them.


**Q7 Justification For Your Score:**

The theoretical results in this paper are sound. Since an earlier paper studies the same problem, it is better to design some examples to further compare the performances of the proposed algorithms in both papers.

**Q9 Complying With Reviewing Instructions:**

1: Yes.

---

### Decision · Program_Chairs · 2022-05-15

**Decision:**

Accept (Poster)

**Comment:**

Meta Review: This paper receives positive reviews from the reviewers. The major merit, the novelty of the proposed method and the soundness of the proof, are in particular appreciated by the reviewers.